# The nasal methylome as a biomarker of asthma and airway inflammation in children

Andres Cardenas [1,2,11], Joanne E. Sordillo[2,11], Sheryl L. Rifas-Shiman[2], Wonil Chung [3], Liming Liang[3], Brent A. Coull[3], Marie-France Hivert[2,4], Peggy S. Lai[5], Erick Forno[6], Juan C. Celedón[6], Augusto A. Litonjua [7], Kasey J. Brennan[8], Dawn L. DeMeo[9], Andrea A. Baccarelli[8], Emily Oken [2] & Diane R. Gold[9,10]

The nasal cellular epigenome may serve as biomarker of airway disease and environmental response. Here we collect nasal swabs from the anterior nares of 547 children (mean-age 12.9 y), and measure DNA methylation (DNAm) with the Infinium MethylationEPIC Bead-Chip. We perform nasal Epigenome-Wide Association analyses (EWAS) of current asthma, allergen sensitization, allergic rhinitis, fractional exhaled nitric oxide (FeNO) and lung function. We find multiple differentially methylated CpGs (FDR < 0.05) and Regions (DMRs; ≥ 5-CpGs and FDR < 0.05) for asthma (285-CpGs), FeNO (8,372-CpGs; 191-DMRs), total IgE (3-CpGs; 3-DMRs), environment IgE (17-CpGs; 4-DMRs), allergic asthma (1,235-CpGs; 7-DMRs) and bronchodilator response (130-CpGs). Discovered DMRs annotated to genes implicated in allergic asthma, Th2 activation and eosinophilia (EPX, IL4, IL13) and genes previously associated with asthma and IgE in EWAS of blood (ACOT7, SLC25A25). Asthma, IgE and FeNO were associated with nasal epigenetic age acceleration. The nasal epigenome is a sensitive biomarker of asthma, allergy and airway inflammation.

[1] Division of Environmental Health Sciences, School of Public Health, University of California, Berkeley, Berkeley, CA 94720, USA. [2] Department of Population Medicine, Division of Chronic Disease Research Across the Lifecourse, Harvard Medical School and Harvard Pilgrim Health Care Institute, Boston, MA 02215, USA. [3] Department of Biostatistics, Harvard T.H. Chan School of Public Health, Boston, MA 02115, USA. [4] Diabetes Unit, Massachusetts General Hospital, Boston 02114 MA, USA. [5] Massachusetts General Hospital, Pulmonary/Critical Care, Boston, MA 02114, USA. [6] Division of Pediatric Pulmonary Medicine, University of Pittsburgh School of Medicine, Pittsburgh, PA 15224, USA. [7] Division of Pediatric Pulmonary Medicine, University of Rochester Medical Center, Rochester, NY 14642, USA. [8] Department of Environmental Health Sciences, Mailman School of Public Health, Columbia University, New York, NY 10032, USA. [9] Department of Medicine, Brigham and Women's Hospital, Channing Division of Network Medicine, Harvard Medical School, Boston, MA 02115, USA. [10] Department of Environmental Health, Harvard T. H. Chan School of Public Health, Boston, MA 02115, USA. [11]These authors contributed equally: Andres Cardenas, Joanne E. Sordillo. Correspondence and requests for materials should be addressed to A.C. (email: andres.cardenas@berkeley.edu)

Asthma, a chronic respiratory disease characterized by reversible airflow obstruction and airway inflammation, affects over 300 million people worldwide[1]. While genetic factors are thought to account for approximately 60% of asthma susceptibility[2], the remaining proportion is believed to result from environmental exposures. The epigenome is at the intersection of these two broad classes of causal factors: genes and environment, providing a unique opportunity to understand the interplay between genetic and environmental factors.

DNA methylation (DNAm) of cytosine nucleotides (CpG sites) is one of the most widely studied epigenetic modifications[3]. DNAm has been shown to be a promising biomarker of diseases[4] and immune phenotypes, including asthma[5,6], allergy[7], and immunoglobulin E (IgE) levels[8,9]. However, most studies have been performed in whole blood DNA. DNAm is tissue specific, and collection of target tissue is critical for the development of disease-relevant biomarkers. The upper respiratory tract is in direct contact with the external environment and respiratory epithelial cells provide multiple barrier functions, including the capability to directly signal immune cells for protection and clearance of microbial pathogens, particles, and toxicants[10,11]. Epithelial barrier abnormalities and unbalanced immune system activation can lead to airway disease. Previous studies have suggested the relevance of nasal cells as epigenetic biomarkers of lower airway disease and asthma[12,13].

In epigenome-wide association analyses (EWAS) of nasal samples collected from 547 early-teen children, we hypothesized that the DNAm signatures would serve as biomarkers of airway disease, airway inflammation, allergen sensitization, and lung function. We theorized that these signatures would reflect epigenetic modifications of T-helper type 2 (Th2) immune signaling and epithelial barrier genes, which are known to play a role in allergy and asthma[14]. In addition, we hypothesized that asthma, allergy, IgE, and lung function would be associated with accelerated epigenetic aging of nasal cells. We show that allergic asthma and elevated biomarkers of allergic disease, such as fractional exhaled nitric oxide (FeNO) and IgE, are associated with multiple differentially methylated CpGs and regions of genes that may alter the structure and function of epithelial cells and genes implicated in allergic asthma, Th2 activation, and eosinophilia. No associations between nasal DNAm and lung function or allergic rhinitis was observed. Finally, asthma and elevated markers of allergic disease are associated with accelerated epigenetic aging in nasal cells.

## Results

**Study characteristics**. We collected nasal DNA samples year round: 33.6% in summer, 21.9% in fall, 19% in winter, and 25.4% in spring. We measured nasal DNAm among $N = 547$ Project Viva participants at the early teen visit with a mean age 12.9 years (SD = 0.65), range 11.9–15.3 years. Participants were 50.6% male, and 67.1% white, 16.1% black, 4.2% Hispanic, 3.1% Asian, and 9.3% of more than one race. Overall, 12% reported current asthma and 16.7% reported symptoms consistent with current allergic rhinitis. Of the 366 participants with IgE sensitization testing, 58.7% were sensitized to environmental allergens (Table 1).

**Global DNAm variability**. Univariate associations of principal components (PCs) from the nasal DNAm data (PCs 1–30 on the $x$-axis) with covariates and surrogates of cell-type mixture (PCs 1–10 from ReFACTor on $y$-axis) are shown in Fig. 1. Overall, the first 30 PCs, derived from 719,075 high-quality CpGs, explained 59% of the variance of the nasal methylome. As expected, cell-type PCs estimated via ReFACTor (a reference-free method)

**Table 1 Study characteristics of Project Viva participants[a]**

| Characteristics | N (%) or mean (SD) |
|---|---|
| Sex | |
| Male | 277 (50.6%) |
| Female | 270 (49.4%) |
| Race/ethnicity | |
| White | 367 (67.1%) |
| Black | 88 (16.1%) |
| Hispanic | 23 (4.2%) |
| Asian | 17 (3.1%) |
| More than one race | 51 (9.3%) |
| Missing | 1 |
| Mother's education | |
| College graduate | 379 (69.5%) |
| No college education | 166 (30.5%) |
| Missing | 2 |
| Child age (years) | 12.9 (0.65) |
| BMI $z$-score | 0.41 (1.07) |
| Missing | 1 |
| Smokers living at home | |
| Yes | 67 (12.4%) |
| No | 474 (87.6%) |
| Missing | 6 |
| Season at sample collection | |
| Summer | 184 (33.6%) |
| Fall | 120 (21.9%) |
| Winter | 104 (19.0%) |
| Spring | 139 (25.4%) |
| Asthma | |
| Current | 65 (12.0%) |
| Past but not current | 74 (13.7%) |
| Never | 402 (73.5%) |
| Missing | 6 |
| Allergic rhinitis | |
| Current | 47 (16.7%) |
| Never | 234 (83.3%) |
| Missing | 266 |
| Environment IgE sensitization | |
| Yes | 215 (58.7%) |
| No | 151 (41.3%) |
| Missing | 181 |
| Total IgE (kU L$^{-1}$) | 189.3 (368.7) |
| Missing | 181 |
| FeNO (ppb) | 27.7 (29.5) |
| Missing | 21 |
| FEV $z$-score | −0.15 (1.05) |
| Missing | 14 |
| FVC $z$-score | 0.07 (1.00) |
| Missing | 14 |
| FEV$_1$/FVC $z$-score | −0.38 (0.99) |
| Missing | 181 |
| BDR | 3.4% (6.8) |
| Missing | 85 |

*BMI* body mass index, *IgE* immunoglobulin E, *FeNO* fractional exhaled nitric oxide, *BDR* bronchodilator response, *FEV* forced expiratory volume, *FVC* forced vital capacity
[a]During the early-teen visit among participants with complete nasal DNA methylation data after quality control ($N = 547$)

showed the strongest associations with top nasal DNAm PCs. Sex, race, age at nasal sample collection, and season were associated with the first nasal DNAm PC. FeNO and total IgE were associated with the second PC suggesting strong associations with these phenotypes.

**CpG-by-CpG and regional DNAm analyses**. In linear models adjusted for child race/ethnicity, sex, age at nasal sample collection, body mass index (BMI) $z$-score, maternal education, smokers living in the household, sine and cosine of season at sample

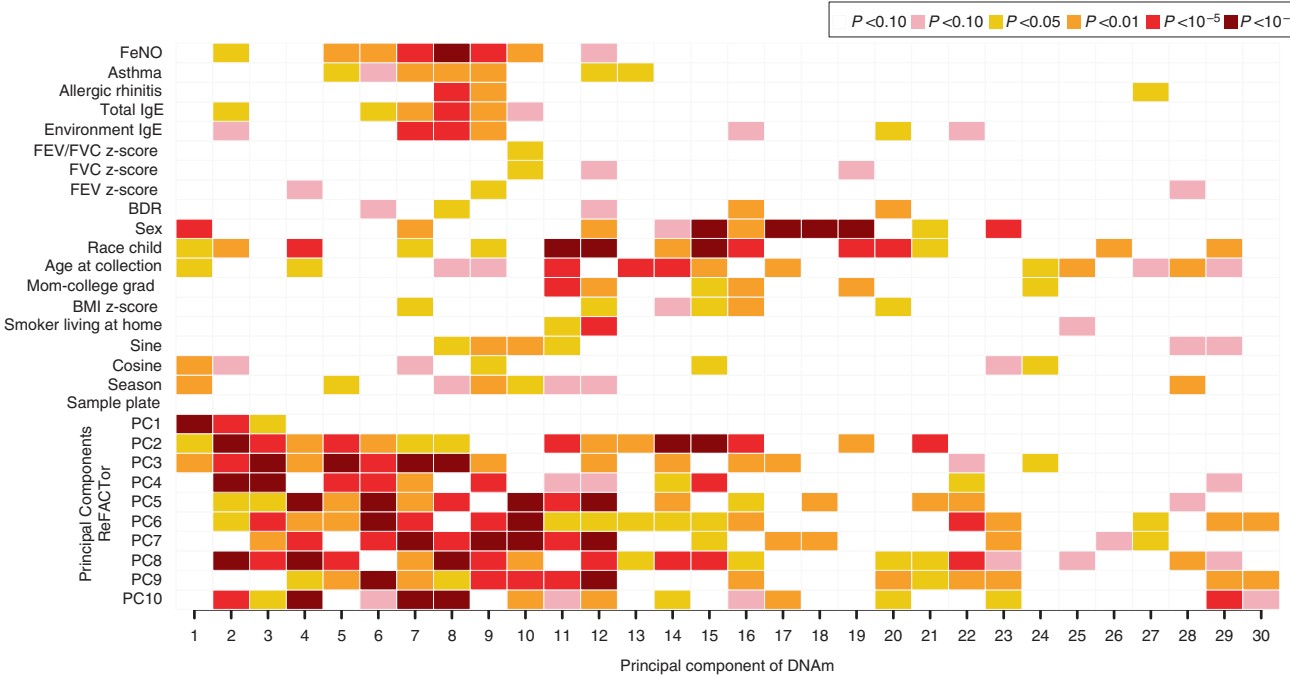

**Fig. 1** Associations with Global Nasal DNA methylation (DNAm) variability. Principal component (PC) regression analysis showing univariate association $P$ values color-coded by smallest $P$ value (dark red; $P < 10^{-10}$) to largest (blank; $P > 0.10$) between all covariates of interest and the top 30 PCs explaining 59% of the variance of the nasal DNA methylation data in the x-axis. PCs 1–10 on the y-axis reflect DNAm heterogeneity and differences in cell types bioinformatically estimated using ReFACTor

### Table 2 Summary of nasal EWAS[a]

| EWAS[b] | Sample size ($N$; $n$ = cases) | Fully adjusted EWAS | | | |
|---|---|---|---|---|---|
| | | $\lambda$ | FDR < 0.05 CpGs | Bonferroni CpGs | DMRs |
| Current asthma vs. never | $N = 463$; $n = 65$ | 0.91 | 285 | 1 | 0 |
| FeNO | $N = 517$ | 1.32 | 8372 | 744 | 191 |
| Current allergic rhinitis vs. never | $N = 277$; $n = 47$ | 1.08 | 0 | 0 | 0 |
| Total IgE | $N = 361$ | 1.09 | 3 | 1 | 3 |
| Environment IgE sensitization | $N = 361$; $n = 211$ | 0.93 | 17 | 6 | 4 |
| Current asthma with environment IgE sensitization | $N = 301$; $n = 36$ | 1.00 | 1235 | 15 | 7 |
| FEV $z$-score | $N = 525$ | 0.99 | 0 | 0 | 0 |
| FVC $z$-score | $N = 525$ | 0.95 | 0 | 0 | 0 |
| FEV/FVC $z$-score | $N = 525$ | 1.11 | 0 | 0 | 0 |
| % Bronchodilator response | $N = 454$ | 0.87 | 130 | 50 | 0 |

$\lambda$ genomic inflation factor, *EWAS* epigenome-wide association analyses, *FeNO* fractional exhaled nitric oxide, *FDR* false discovery rate, *DMRs* differentially methylated regions with ≥5 CpGs with a Stouffer FDR $P < 0.05$, *IgE* immunoglobulin E, *FEV* forced expiratory volume, *FVC* forced vital capacity, *PCs* principal components
[a]Number of differentially methylated CpGs (FDR <0.05 and Bonferroni correction) and DMRs (≥5-CpGs and FDR < 0.05)
[b]Analyses adjusted for child race/ethnicity, sex, age at sample collection, BMI $z$-score, maternal education, smokers living in the house, sine and cosine of season at sample collection, and cellular heterogeneity (10 PCs from ReFACToR)

collection, and cell-type heterogeneity, we observed multiple differentially methylated CpGs (false discovery rate (FDR) < 0.05) and regions (DMRs; ≥5-CpGs and FDR < 0.05) for current asthma (285 CpGs), FeNO (8372 CpGs; 191 DMRs), total IgE (3 CpGs; 3 DMRs), environmental allergen IgE sensitization (17 CpGs; 4 DMRs), allergic asthma with IgE sensitization (1235 CpGs; 7 DMRs), and bronchodilator response (BDR) (130 CpGs). Allergic rhinitis and lung function measures (forced expiratory volume (FEV), forced vital capacity (FVC), FEV/FVC ratio) were not associated with differential DNAm of nasal cells in adjusted models (Table 2). Genomic inflation observed in EWAS not adjusted for cell-type heterogeneity was greatly attenuated after adjustment for surrogates of cell types estimated via ReFACTor (Supplementary Figure 1). Manhattan plots of fully adjusted

EWAS of (a) FeNO, (b) current asthma, and (c) allergic asthma are shown in Fig. 2.

Biological pathway analyses using the Kyoto Encyclopedia of Genes and Genomes (KEGG) showed that the asthma pathway was differentially methylated among significant genes in EWAS of allergic asthma and FeNO. In addition, the interleukin-17 (IL-17) signaling pathway was differentially methylated for FeNO (Supplementary Table 1).

**Overlap across phenotypes**. By far, the largest number of differentially methylated sites and DMRs were associated with FeNO levels. The majority of differentially methylated CpGs (FDR < 0.05) associated with current asthma, total IgE, positive

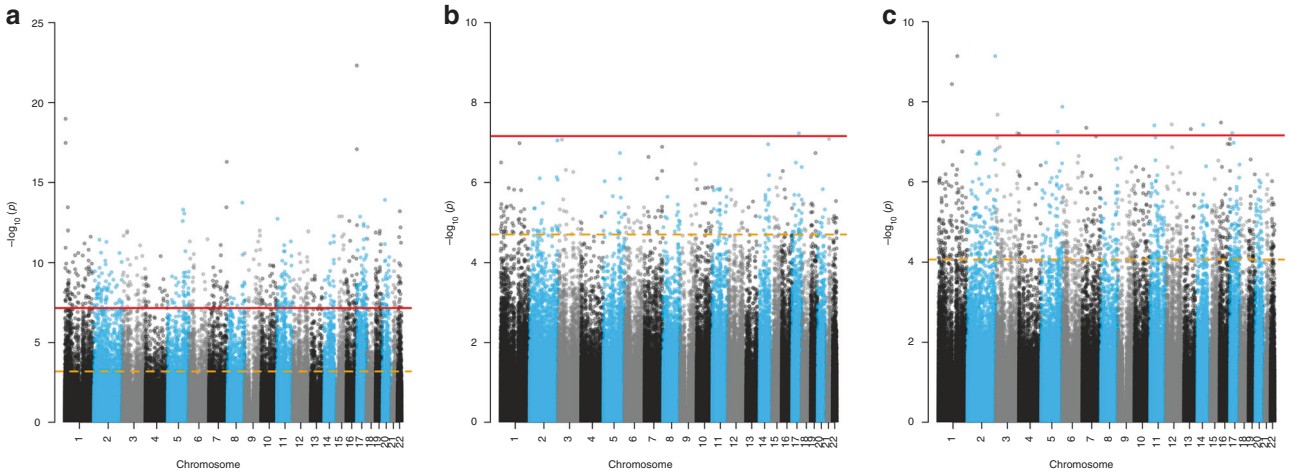

**Fig. 2** Nasal epigenome-wide associations for fractional exhaled nitric oxide (FeNO), asthma, and allergic asthma. Uncorrected $-\log_{10}(P)$ values) plotted in the y-axis for each CpG site and sorted by chromosomal and genomic position on the x-axis. Adjustment for multiple testing was accounted for each epigenome-wide association analyses by controlling the false discovery rate (FDR <0.05), horizontal orange dashed line, and Bonferroni threshold for statistical significance ($P < 6.95 \times 10^{-8}$) shown in the solid red horizontal line. **a** Manhattan plot of epigenome-wide association analyses (EWAS) for fractional exhaled nitric oxide (FeNO), **b** Manhattan plot of EWAS for current asthma, and **c** Manhattan plot of EWAS for current allergic asthma

**Table 3 Top 20 differentially methylated CpGs associated with current asthma compared to never reporting asthma**

| CpG | Chr | % difference in DNAm | Unadjusted P | q value | UCSC gene | Other close gene | Region |
|-----|-----|----------------------|--------------|---------|-----------|------------------|--------|
| cg10555106 | 1 | −3.20 | 3.19E − 07 | 0.019 | LOC388588 | SMIM1 | Inside |
| cg04165922 | 1 | 5.24 | 1.05E − 07 | 0.013 | NOS1AP | NOS1AP | Inside |
| cg05461268 | 2 | 6.09 | 7.45E − 07 | 0.024 | DNER | DNER | Inside |
| cg10117579 | 2 | 3.57 | 8.85E − 08 | 0.013 | | COPS8 | Upstream |
| cg24113459 | 3 | −2.65 | 8.48E − 08 | 0.013 | GLB1 | GLB1 | Inside |
| cg04217850 | 3 | −1.82 | 4.89E − 07 | 0.022 | SLC25A26 | SLC25A26 | Inside |
| cg13521315 | 3 | 2.43 | 5.52E − 07 | 0.022 | TMCC1 | TMCC1 | Inside |
| cg27187848 | 5 | 2.84 | 1.84E − 07 | 0.015 | PCDHGA8 | PCDHGA8 | Inside |
| cg18321881 | 7 | 3.42 | 2.31E − 07 | 0.017 | FKBP9 | AVL9 | Inside |
| cg18666454 | 7 | −2.71 | 5.13E − 07 | 0.022 | KCNH2 | KCNH2 | Inside |
| cg22060869 | 7 | −1.95 | 1.29E − 07 | 0.013 | KCNH2 | KCNH2 | Inside |
| cg16044211 | 9 | 0.48 | 3.42E − 07 | 0.019 | FAM129B | FAM129B | Inside |
| cg11157850 | 14 | −2.25 | 1.11E − 07 | 0.013 | | CHGA | Upstream |
| cg16409452 | 14 | −2.34 | 6.59E − 07 | 0.024 | EVL | EVL | Inside |
| cg17148519 | 16 | −1.26 | 5.18E − 07 | 0.022 | USP7 | USP7 | Inside |
| cg04010582 | 17 | −1.51 | 3.21E − 07 | 0.019 | RAB11FIP4 | RAB11FIP4 | Inside |
| cg04727951 | 17 | −2.00 | 5.94E − 08 | 0.013 | MSI2 | MSI2 | Inside |
| cg13000015 | 17 | −2.07 | 4.14E − 07 | 0.021 | CCDC57 | CCDC57 | Inside |
| cg17054995 | 19 | −1.45 | 1.84E − 07 | 0.015 | LGALS4 | LGALS4 | Promoter |
| cg24082440 | 21 | −1.06 | 8.16E − 08 | 0.013 | MRAP | MRAP | Inside |

*Chr* chromosome, *DNAm* DNA methylation

specific IgE to environmental allergens and allergic asthma were a subset of the sites associated with FeNO (>79%) (Supplementary Table 2). The top 20 differentially methylated CpGs ranked on significance associated with current asthma are shown in Table 3. In addition, multiple DMRs with five or more CpGs (*EPX* (eosinophil peroxidase), *ACOT7*, *SORCS2* genes) were overlapping across phenotypes like FeNO, allergic asthma, environmental IgE sensitization, and total IgE (Tables 4 and 5). BDR was the only trait with a unique nasal DNAm signature, showing no overlap with any of the other phenotypes considered (Supplementary Table 2). Summary results and CpG annotation for significant EWAS results are shown in the supplementary material for Asthma (Supplementary Data 1), FeNo (Supplementary Data 2), total IgE (Supplementary Table 3), Environment IgE sensitization (Supplementary Table 4), allergic asthma (Supplementary Data 3), and BDR (Supplementary Data 4).

Functional gene categories represented in DMRs and individual CpGs found for FeNO, asthma, and allergy were eosinophilic activity (*EPX*, *CLC*, *PRG2*), and Th2 responses (*IL-4*, *ZFPM1*). Differentially methylated CpGs and sites associated with FeNO related to functions of solute transport and intracellular membrane trafficking (*SLC25A25*, *SLC39A4*, *DNAH17*, *VTI1A*), T cell activation (*LAX*), oxidative stress (*VKORC1L1*), and mucin production (*GALNT7*). We also observed differential nasal DNAm of asthma-associated genes identified in previous independent genome-wide association studies (GWAS): *TNIP-1*[15], *IL-13*[16], and *CHI3L1*[17] for FeNO DMRs (Supplementary Data 5). Lower DNAm of the *CHI3L1* gene responsible for the production of the chitinase-like protein YKL-40, a mediator associated with Th2 responses[18], was differentially methylated for a DMR relative to FeNO and for a CpG (cg19081101) for asthma and allergic asthma. Greater DNAm of several CpGs annotated to the *PRTN3* gene was associated with higher FeNO and allergic

**Table 4 Top 20 DMRs associated with $\log_{10}$ FeNO measurements**

| Chromosome: Position | Width (bp) | Number of CpGs | Stouffer P | Mean % difference in DNAm | UCSC gene |
|---|---|---|---|---|---|
| Chr17:56,269,170–56,270,828 | 1659 | 8 | 6.85E − 40 | −3.08 | EPX |
| Chr11:44,091,987–44,092,987 | 1001 | 5 | 9.28E − 26 | −2.18 | ACCS |
| Chr1:6,340,110–6,342,888 | 2779 | 9 | 2.57E − 21 | −2.84 | ACOT7 |
| Chr19:847,943–848,896 | 954 | 5 | 4.09E − 19 | 2.92 | PRTN3 |
| Chr17:62,307,787–62,309,293 | 1507 | 7 | 2.02E − 18 | −1.88 | TEX2 |
| Chr1:203,155,884–203,156,784 | 901 | 6 | 7.31E − 18 | 2.51 | CHI3L1 |
| Chr10:114,437,411–114,438,072 | 662 | 5 | 1.35E − 17 | −1.71 | VTI1A |
| Chr16:88,539,861–88,540,992 | 1132 | 7 | 2.90E − 16 | −2.52 | ZFPM1 |
| Chr16:357,249–358,591 | 1343 | 7 | 4.78E − 15 | −1.63 | AXIN1 |
| Chr1:27,240,319–27,241,913 | 1595 | 11 | 1.31E − 14 | −2.12 | NR0B2 |
| Chr17:81,039,326–81,043,782 | 4457 | 17 | 2.67E − 13 | −2.02 | METRNL |
| Chr11:64,642,144–64,643,272 | 1129 | 7 | 3.01E − 12 | −1.29 | EHD1 |
| Chr17:72,442,179–72,443,401 | 1223 | 6 | 3.38E − 12 | 3.14 | GPRC5C |
| Chr20:35,503,983–35,504,553 | 571 | 8 | 1.20E − 11 | −2.26 | TLDC2 |
| Chr4:7,637,289–7,638,050 | 762 | 5 | 5.54E − 11 | 2.43 | SORCS2 |
| Chr20:4,764,077–4,764,312 | 236 | 5 | 5.57E − 11 | −1.19 | RASSF2 |
| Chr17:76,564,134–76,565,673 | 1540 | 8 | 8.00E − 11 | −1.66 | DNAH17 |
| Chr8:141,046,436–141,047,408 | 973 | 6 | 1.10E − 10 | −1.85 | TRAPPC9 |
| Chr5:132,008,525–132,009,947 | 1423 | 7 | 2.71E − 10 | −1.73 | IL4 |
| Chr9:130,859,191–130,860,839 | 1649 | 11 | 1.11E − 09 | −1.21 | SLC25A25 |

*DMRs* differentially methylated regions, *FeNO* fractional exhaled nitric oxide, *DNAm* DNA methylation

**Table 5 DMRs associated with $\log_{10}$ total IgE, environment IgE sensitization, and allergic asthma**

| Chromosome: Position | DMRs for total IgE | | | | |
|---|---|---|---|---|---|
| | Width (bp) | Number of CpGs | Stouffer P | Mean % difference in DNAm | UCSC gene |
| Chr17:56,269,170–56,270,249 | 1080 | 7 | 0.019 | −1.01 | EPX |
| Chr4:7,637,289–7,638,050 | 762 | 5 | 0.025 | 0.95 | SORCS2 |
| Chr6:41,130,718–41,131,213 | 496 | 5 | 0.029 | 0.85 | TREM2 |
| | DMRs for environment IgE sensitization | | | | |
| Chr4:7,637,289–7,638,050 | 762 | 5 | 0.006 | 1.32 | SORCS2 |
| Chr11:73,115,417–73,116,329 | 913 | 5 | 0.010 | 1.07 | FAM168A |
| Chr19:1,154,276–1,155,738 | 1463 | 6 | 0.019 | 1.07 | SBNO2 |
| Chr1:6,340,443–6,342,159 | 1717 | 6 | 0.027 | −1.31 | ACOT7 |
| | DMRs for allergic asthma | | | | |
| Chr17:56,269,170–56,270,828 | 1659 | 8 | 1.82E − 04 | −2.87 | EPX |
| Chr1:87,596,049–87,597,369 | 1321 | 5 | 4.12E − 04 | −2.59 | LINC01140 |
| Chr20:4,764,077–4,764,312 | 236 | 5 | 9.54E − 04 | −1.58 | RASSF2 |
| Chr6:29,592,854–29,593,913 | 1060 | 5 | 0.027 | 3.10 | GABBR1 |
| Chr1:6,645,094–66,45,463 | 370 | 5 | 0.030 | −2.60 | ZBTB48 |
| Chr9:116,326,652–116,327,278 | 627 | 6 | 0.033 | 3.67 | RGS3 |
| Chr9:101,705,161–101,705,792 | 632 | 5 | 0.033 | −1.58 | COL15A1 |

*DMRs* differentially methylated regions, *IgE* immunoglobulin E, *DNAm* DNA methylation

asthma. The PRTN3 protein has been previously shown to have altered levels of abundance in the nasal epithelium of individuals with current allergic rhinitis[19].

In analyses of asthma and allergy, we observed lower DNAm of several CpGs in genes regulating eosinophilic and Th2 responses: *EPX* and *IL-4*, respectively. Lower DNAm of *PRG2*, which encodes for a pro-eosinophil major basic protein, was associated with FeNO and allergic asthma. Lower DNAm of *CLC*, a gene for lysophospholipase expressed in eosinophils, was associated with current asthma, allergic asthma, and FeNO. The *ZFPM1* gene known to facilitate Th1 differentiation through the downregulation of the Th2 cytokine IL-4[20] had lower DNAm levels for participants with higher levels of FeNO (DMR), total IgE, environment IgE sensitization, asthma, and allergic asthma.

Solute carriers and intracellular transport genes were differentially methylated. For example, the *VTI1A* gene, part of the SNARE protein family associated with membrane transport and permeability, had lower DNAm for one site (cg26724455) among allergic asthmatics and was differentially methylated for several sites relative to higher FeNO. Lower regional DNAm in genes for solute carriers and intracellular transport (*SLC25A25*, *DNAH17*) were associated with FeNO. In addition, a DMR in the *ADAM-8* gene, a metalloprotease involved in cell matrix interactions, was hypomethylated for FeNO (Supplementary Data 5). Another DMR associated with FeNO and environment IgE sensitization was found within *SBNO2*[21], a downstream mediator of anti-inflammatory IL-10 responses. Hypomethylation of a gap junction protein gene (*GJA4*) was observed for sensitization to environmental allergens (Supplementary Table 4).

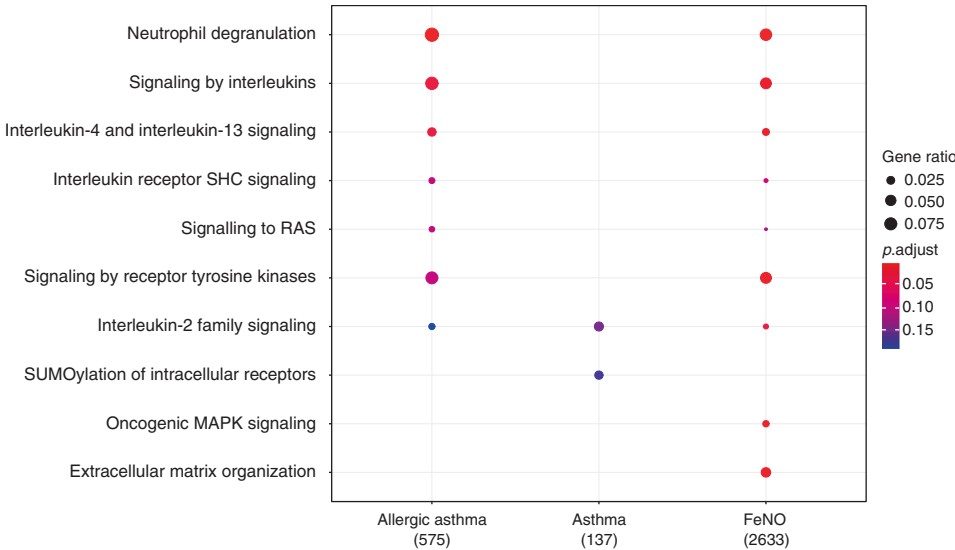

**Fig. 3** Differentially methylated gene ontology categories. Gene enrichment analyses for gene ontologies present in the Reactome database that were differentially methylated in epigenome-wide association analyses (false discovery rate (FDR) < 0.05) for allergic asthma, asthma, and fractional exhaled nitric oxide (FeNO)

Among differentially methylated genes, we also looked at biological pathways enriched in the Reactome database. The top differentially methylated Reactome pathway was observed for neutrophil degranulation followed by signaling by interleukin pathways for allergic asthma and FeNO. Genes found for asthmatics were enriched in the interleukin-2 family signaling as well as the SUMOylation of intracellular receptor pathways (Fig. 3).

**Previous findings of asthma and blood DNAm**. In addition to identifying differentially methylated CpGs and DMRs from our nasal samples, we checked individual CpGs found in whole blood to be associated with asthma in children from a large meta-analysis[5]. This meta-analysis identified 14 differentially methylated CpGs in leukocytes from children with asthma compared to non-asthmatics using data from six European cohorts (age 4–16 years). In our nasal DNAm data, 13 sites were differentially methylated and consistent in direction. We were unable to test one of the CpGs (cg11456013; *AMD1*) as it is unique to the Infinium Human Methylation 450K (Supplementary Table 5). One of the previously reported CpGs (cg13835688) in this meta-analysis was annotated to the *SLC25A25* gene. In our agnostic regional analyses, we observed an entire DMR with lower DNAm covering 11 CpGs for the *SLC25A25* gene associated with higher FeNO.

**Replication in nasal epithelial cells**. We tested for replication of our top differentially methylated findings of asthma and allergic asthma in an external cohort with nasal epithelial cells collected from the posterior portion of the inferior turbinate and analyzed with the Infinium Human Methylation 450K BeadChip[6]. We checked for replication of the 285 differentially methylated CpGs for asthmatics of which 95 probes were found in the 450K. Among the 95 CpGs found to be differentially methylated for asthmatics in our study and present in the 450K, 58 CpGs replicated (61%) after controlling for multiple comparisons (FDR < 0.05 for 95 comparisons) in the replication cohort with consistent direction and magnitude of association for all 58 CpGs (Supplementary Data 6). Replicated CpGs included several sites annotated to the *ACOT7*, *EPX*, and *EVL* genes. For allergic asthma analyses, there were 375 CpG probes present in the Yang

et al. 450K dataset of which 199 CpGs (53%) replicated with an FDR < 0.05 and 197 of these had consistent direction of association (Supplementary Data 7). Replicated CpGs included several sites annotated to the *ACOT7*, *ZFPM1*, *PRG2*, *EPX*, and *EVL* genes.

We also compared our results to data from EVA-PR, a case–control study of asthma in Puerto Rican children and adolescents, which measured the nasal methylome using nasal epithelial cells collected from the inferior turbinate[22]. In this external cohort's findings for atopic asthma, we identified 61% (48/79) of CpGs that replicated (FDR < 0.05 for 48 comparisons), all with consistent direction of association including multiple CpGs annotated to *EVL* and *EPX* genes for our asthma results (Supplementary Data 6). For our allergic asthma results, 59% (187/315) of CpGs replicated (FDR < 0.05 for 187 comparisons), but two associations had opposite directions of effect (Supplementary Data 7). Lastly, for environment IgE sensitization four out seven CpGs replicated (*GJA4*, *ACOX2*, *PRKAG2*, *CYP27B1/ METTL1*) and all seven associations had consistent directions of effect (Supplemental Table 6). Notably, in the EVA-PR study analysis of atopy, two of our top CpGs were replicated among the top 30 differentially methylated findings from sorted CD326+ epithelial cells (cg15006973; *GJA4* and cg20372759; *CYP27B1/ METTL1* genes). In this cohort, they were able to test for DNAm and gene expression relationships, and several of our top differentially methylated CpGs were shown to be associated with expression. Namely, for environment IgE sensitization (*METTL1*), for allergic asthma (*NTRK1*), and several for FeNO (*MAP3K14*, *NTRK1*, *FBXL7*, *PCSK6*, *SLC9A3*, *CDH26*, *CAPN14*, and *MAP3K14*).

**Epigenetic clock: epigenetic age acceleration**. We evaluated epigenetic age acceleration estimated using the nasal methylome. The correlation between chronological age and DNAm-Age was moderate ($r = 0.19$, $P < 0.0001$, Supplementary Fig. 2). We observed significant epigenetic age acceleration in children with current asthma (0.74 years; 95% confidence interval (CI): 0.02, 1.46) and even greater age acceleration for children with allergic asthma (1.30 years; 95% CI: 0.36, 2.23). For every 10-fold increase in FeNO, we observed that epigenetic age was accelerated by 1.11 years (95% CI: 0.39, 1.83). Similarly, every 10-fold increase in

**Table 6 Association of epigenetic age acceleration with asthma, allergy, and lung function**

| Predictor | Epigenetic age acceleration[a] (95% CI) | P value |
|---|---|---|
| Current asthma | 0.74 (0.02, 1.46) | 0.04 |
| FeNO (ppb) | 1.11 (0.39, 1.83) | 0.003 |
| Current allergic rhinitis | 0.61 (−0.25, 1.46) | 0.16 |
| Total IgE (kU L$^{-1}$) | 0.58 (0.15, 1.01) | 0.009 |
| Environment specific IgE sensitization | 0.83 (0.27, 1.39) | 0.004 |
| Current asthma with environment IgE sensitization | 1.30 (0.36, 2.23) | 0.007 |
| FEV z-score | −0.13 (−0.36, 0.09) | 0.24 |
| FVC z-score | −0.09 (−0.32, 0.15) | 0.47 |
| FEV/FVC z-score | −0.11 (−0.35, 0.13) | 0.35 |
| % Bronchodilator response | 0.01 (−0.03, 0.05) | 0.59 |

*FeNO* fractional exhaled nitric oxide, *CI* confidence interval, *IgE* immunoglobulin E, *FEV* forced expiratory volume, *FVC* forced vital capacity, *DNAm* DNA methylation
[a]Residuals ($\varepsilon_i$) of DNAm-Age −chronological age + $\varepsilon_i$

total IgE was associated with epigenetic age acceleration of 0.58 years (95% CI: 0.15, 1.01). Participants with environmental IgE sensitization had on average 0.83 years of epigenetic age acceleration (95% CI: 0.27, 1.39) (Table 6).

## Discussion

We report epigenome-wide associations for nasal samples among several traits related to asthma, allergy, and lung function. For current asthma and elevated biomarkers of allergic disease (FeNO and IgE), we observed multiple DMRs of genes that may alter the structure and function of epithelial cells, including those encoding for solute carriers and membrane transport proteins, oxidative stress response and mucin production enzymes. We observed a strong, consistent signal for lower DNAm of genes driving Th2 and eosinophilic responses, with *EPX* as the top DMR across asthma and allergy phenotypes. We did not observe associations between nasal DNAm and lung function or allergic rhinitis. Finally, we show that epigenetic age of nasal samples is accelerated by asthma and is correlated with elevated biomarkers of allergic disease.

Comparison of our nasal methylome findings to a meta-analysis of asthma in circulating leukocytes[5] from children yielded consistent results. Asthma associated CpG sites tested in whole blood were also differentially methylated in our sample of nasal cells. This consistency suggests that the methylome signature in asthma is stable in whole blood, nasal cells, and potentially other tissues. We observed consistent lower nasal DNAm in asthmatics at the CpGs identified and even greater effect sizes when compared to meta-analysis results from whole blood. In the meta-analysis, the authors also reported on effect sizes from isolated eosinophils of asthmatic vs. non-asthmatics and observed consistent and even stronger associations (>10% DNAm differences) in this target cell type. While our associations are consistent and stronger in magnitude compared to whole blood results, they are weaker (<6% DNAm differences) when compared to the isolated eosinophil results, suggesting that the strength of these epigenetic signatures may vary by target tissue. Eosinophils are present in normal mucosa, but their abundance is greater in the late phase of atopic reaction[23]. While differences in DNAm in whole blood from the EWAS meta-analysis were replicated in isolated eosinophils, none of the differentially methylated genes were directly associated with eosinophil function. In contrast, our nasal methylome DMR findings identified lower DNAm of the *EPX* for FeNO, total IgE, and allergic asthma.

In addition to comparing our results to EWAS in whole blood, we replicated our findings in two independent case–control studies that used the Illumina 450K and sampled nasal epithelial cells from the inferior turbinate[6]. While we were not able to test for replication of all sites due to probe density differences across arrays and preprocessing, we did replicate >50% of the findings from CpGs available in the two cohorts with consistent magnitude of effects, including multiple CpGs annotated to *ACOT7*, *ZFPM1*, *PRG2*, *EPX*, *GJA4 CYP27B1/METTL1*, and *EVL* genes.

Lastly, we observed differential DNAm of genomic regions of three genes previously associated with asthma or biomarkers of asthma in GWAS, *TNIP-1*[15], *IL-13*[16], and *CHI3L1*[17]. These findings illustrate the relevance of nasal anterior nares cells for epigenetic studies of asthma. Two of these genes, *IL-13* and *CHI3L1*, show high expression of their protein products in the lungs of asthmatics, and both are known to regulate Th2 response[24,25]. In contrast, none of the DNAm sites reported in the blood leukocyte EWAS meta-analyses were within known asthma GWAS hits[5]. Lower regional DNAm of the *ADAM-8* gene, an airway remodeling metalloprotease acting downstream[26] of the asthma GWAS gene *ORMDL3*[16,27,28], was observed for FeNO in our data. These results indicate that the nasal cellular compartment may be more sensitive to asthma-associated DNAm differences compared to whole blood, and potentially an optimal tissue for detecting epigenetic modifications of known biologically relevant.

We observed minimal overlap of our nasal DNAm findings with cord blood DNAm patterns prospectively linked to development of childhood asthma and allergy. For example, higher DNAm of *SMAD3* in infant's cord blood DNA has been associated with asthma risk in three independent cohort studies, particularly in children with a maternal history of asthma[29]. This modulator of tumor growth factor-β (TGF-β) signaling, with downstream implications for differentiation of T-regulatory cells and Th17 cells[30], was not differentially methylated in our analysis. However, hypermethylation of *TGFB1* itself was observed in our nasal methylome analysis for allergic asthma and FeNO, suggesting that TGF-β signaling is altered in current disease or with concomitant elevated allergic disease biomarkers.

In Project Viva, our own prior analysis of cord blood DNAm identified several associations with mid-childhood (7.8 years) IgE levels[8]. These longitudinal associations were not re-capitulated in the current cross-sectional analysis. In contrast to the cord blood analyses, many of the genes cross-sectionally associated with IgE in whole blood analyses from mid-childhood (*ACOT7*, *ZFPM1*, *IL4*, *IL5RA*, *EPX*, and *PRG2*)[8] consistently had lower nasal DNAm among DMRs found for FeNO, total IgE (*EPX*), environment IgE (*ACOT7*), and allergic asthma (*EPX*). Epigenetic signatures present at birth with an influence on physiological trajectories may not persist, so that later assessments of epigenome-wide differences by current disease status may differ from the transient epigenetic modifications or might vary by tissue. Our previous analyses of prenatal environmental exposures in cord blood and subsequent child blood suggest that DNAm associations might be malleable and non-persistent as children develop[31,32].

One previous study from the Inner City Asthma Consortium interrogated peripheral blood mononuclear cells (PBMCs) from asthmatic children compared to controls. In this study comparing 97 childhood asthma cases and 97 controls, Yang et al.[33] reported differential PBMC DNAm of genes involved in T cell maturation (*RUNX3*), Th2 immunity (*IL-4*), and oxidative stress (catalase). Some of their gene-specific findings overlap with specific DNAm marks in our data for asthma: FeNO and environment IgE sensitization (lower DNAm of *IL-4*, *ACOT7*, *ZFPM1*), or FeNO and allergic asthma (*RUNX3*); whereas other findings revealed

different DNAm marks, although the functions of the differentially methylated genes were similar to those reported in our findings (oxidative stress, T cell differentiation). An additional nasal epigenome wide study in the Inner City Asthma Consortium compared DNAm and gene expression of the nasal epithelium in 36 atopic asthmatic subjects and 36 controls[6]. Interestingly, top DMRs were not in Th2 pathway genes, but instead identified altered DNAm signatures for genes associated with epithelial cell migration, components of the extracellular matrix, cell adhesion, epigenetic regulation, and airway obstruction. These two studies differ from our own in that both contained smaller sample sizes, participants were from different racial and SES backgrounds, used the older array (450K), and neither study included a nasal EWAS of allergic disease biomarker levels such as FeNO, which in our data showed the greatest number of associations. However, we were able to replicate over 50% our findings in samples of nasal epithelial cells observed to be differentially methylated for asthma, allergic asthma, and IgE sensitization found in the 450K data from two previous studies with different methodologies.

In our EWAS of asthma and related phenotypes, we chose to sample from the anterior nares, as it provides a good proxy of DNAm states in the airway[34], arguably the most relevant target tissue for asthma and allergic airway inflammation. We sampled from the anterior nares, a nasal region shown to have similar within-subject DNAm profiles as compared to cells from the inferior turbinate. Sampling from the anterior nares does not require a speculum, and subjects report less discomfort with this technique as compared to inferior turbinate sampling, making it more amenable for use in pediatric populations[35]. While we did not count cell populations in our nasal swab samples, prior work on the cell distributions of samples from the anterior nares of healthy adult subjects showed, on average, 65% respiratory epithelial cells, approximately 34% squamous cells, and ≤1% of inflammatory cells[35]. Therefore, differences in DNAm by asthma or allergic disease phenotypes can arise from any of these cell types. Indeed, differential DNAm of genes associated with respiratory epithelial barrier integrity as well as immune response were observed in our results. To determine whether our DNAm signatures may simply reflect differences in cell type composition by phenotype, we performed reference-free cell-type adjustment using ReFACTor, a method shown to perform well when benchmarked to reference-based methods[36]. Signals were markedly reduced for asthma, IgE, and environment-specific IgE, suggesting that a large proportion of the variability might be driven by differences in cell types. However, multiple DMRs remained for FeNO, suggesting that some of the DNAm biomarkers are independent of cell-type differences, such as DMRs in *EPX* and *EVL* genes that replicated in two other cohorts.

In addition to identifying disease biomarkers, DNAm also serves as a marker of biological aging. For example, the epigenetic clock has been linked to greater risk of diseases and mortality[37,38]. Our own cohort has reported associations between epigenetic age, estimated in blood, in relationship to asthma, allergy, or lung function[39]. Consistent with this previous finding from blood, we also observed epigenetic age acceleration by asthma and allergy using nasal samples. Not surprisingly, the correlation between chronological age and DNAm-Age was moderate ($r = 0.19$) given the small SD in chronological age for our sample (SD = 0.65 years)[40]. Yet, our results show that nasal samples are a viable tissue to capture epigenetic age acceleration correlated with asthma and allergy.

Our analysis of the nasal methylome in asthma and its related phenotypes has several strengths, namely the size of the study (over 500 subjects), the use of multiple phenotypes related to allergic disease, including asthma, biomarkers of allergy, and

airway inflammation, assessment of over 700,000 CpG sites, and over 50% replication rate in two independent cohorts. Some weaknesses and caveats also deserve mention. We were not able to directly assess cell-type composition. We did however adjust bioinformatically for cell-type heterogeneity. Additionally, while it was our intention to measure gene expression, we were unable to collect sufficient high-quality RNA from the anterior nares or enough concentration (250–300 ng μL$^{-1}$).

We observed widespread nasal DNAm variability associated with asthma, allergic asthma, FeNO, total IgE, environment IgE, and to a lesser extent BDR. Significant DNAm variability included known asthma-associated genes, genes involved in Th2 and eosinophilic responses, and genes that may alter permeability and function of the respiratory epithelium. We validated previous results from blood EWAS for asthma and replicated findings in two independent cohorts with nasal epithelial cells. The DNAm signatures might indeed reflect the cellular milieu serving as a sensitive biomarker of airway disease. Further assessment of this DNAm variability will contribute to asthma endotyping and may help elucidate epigenomic targets for interventions, monitoring and progression to improve the course of airway diseases. Finally, we show that epigenetic age is accelerated with elevated biomarkers of allergy and asthma. Taken together, our findings indicate that the nasal cellular epigenome serves as a sensitive epigenetic biomarker of asthma, allergy, and airway responsiveness, as well as the compartment specific biologic processes underlie these phenotypes.

## Methods

**Study population**. Children were participants in Project Viva, a prospective pre-birth cohort study recruited between 1999 and 2002 during the mothers' first prenatal visits at Atrius Harvard Vanguard Medical Associates, a multispecialty medical group practice in Massachusetts, United States[41]. Eligibility criteria included fluency in English, gestational age <22 weeks at the first prenatal visit, and singleton pregnancy. Of the total 2128 live births, 547 children were re-contacted during an early-teen in-person visit (mean 12.9 years) and provided consent for nasal swab sample collection. Mothers provided written informed consent at recruitment and at postpartum follow-up visits. The Institutional Review Board of Harvard Pilgrim Health Care reviewed and approved all study protocols. Nasal cell DNA collection, fraction exhaled nitric oxide, BDR, allergen sensitization, and the current asthma outcome were all assessed at the same time-point during nasal sample collection.

**DNAm measurements**. Trained research assistants collected nasal swabs from the anterior nares, previously demonstrated to yield respiratory epithelial cells[35]. Nasal swabs were immediately stored in lysis buffer and frozen until processing. DNA was isolated using the Maxwell 16 Buccal Swab LEV DNA Purification Kit following the manufacturer's instructions (Promega, Madison, WI, USA).

Epigenome-wide DNAm measurements were performed on DNA extracted from nasal samples with the Infinium MethylationEPIC BeadChip (Illumina, San Diego, CA, USA). The Infinium MethylationEPIC BeadChip quantifies DNAm in over 850,000 CpGs at a single-nucleotide resolution for each sample. Samples were randomly allocated to seven 96-well plates. Sample plates and chips were randomized to ensure balance by sex, current asthma status, current allergic rhinitis, and race to minimize potential confounding by batch effects.

DNAm data were imported into the R statistical software for preprocessing using *minfi*[42]. We first performed quality control at the sample level, excluding samples with overall low intensities, which indicates low-quality (intensities < 10.5; $n = 3$), samples that mismatched on recorded sex ($n = 4$) and samples with mixed genotype distributions on the measured SNP probes (59 SNP probes), indicating possible sample contamination ($n = 8$). In addition, we excluded technical duplicates ($n = 35$). A total of 547 high-quality samples were retained for analyses.

We performed quality control at the probe level by computing a detection $P$ value relative to control probes. Namely, we excluded 4161 probes with non-significant detection ($P > 0.05$) for 5% or more of the samples. We also excluded 18,978 probes annotated to sex chromosomes, 2835 non-CpG probes, 5516 probes with SNPs at the single base extension (minor allele frequency (MAF) ≥ 5%), 70,737 probes containing an SNP (MAF ≥ 5%), and 5215 probes with an SNP at the CpG site (MAF ≥ 5%). Finally, we excluded 40,377 cross-reactive probes previously identified in the MethylationEPIC BeadChip[43]. Subsequently, a total of 719,075 high-quality probes were included in all analyses (Supplementary Table 7). We preprocessed our data using functional normalization with three PCs from the control probes to adjust for technical variability[44]. To adjust for probe-type bias (I vs. II), we used the regression on correlated probes method that leverages

genomic proximity to adjust the distribution of type 2 probes[45]. Lastly, we used ComBat from the *sva* package to adjust for sample plate (seven 96-well plates) as a technical batch variable[46]. We visualized the data using density distributions at all processing steps and performed PC analyses to examine the associations of methylation differences with technical, biological, and measured traits with global DNAm variation using ENmix[47], and PCA plots for technical duplicate agreement.

**Fractional exhaled nitric oxide**. NO is a mediator involved in chronic inflammatory diseases and Th2-mediated immune responses.[48] Measurement of NO in exhaled air, as FeNO, is a non-invasive biomarker of airway inflammation that correlates with airway eosinophilia.[49] We measured FeNO levels with a portable electrochemical device (NIOX MINO; Aerocrine AB, Stockholm, Sweden), which is in agreement with published procedures for FeNO measurement.[50] Prior to measurement of FeNO, we asked participants to breath in through an NO scrubbing filter and then exhale out into the room air two times. We then instructed participants to inhale through the scrubbing filter and exhale into the FeNO analyzer at a flow rate of 50 mL s$^{-1}$. Following American Thoracic Society guidelines, to ensure measurement of lower, not upper, airway FeNO, we did not use nose clips and we quantified the last 3 s of the exhalation.[50] We performed this procedure twice. The mean of these two measurements was used in all subsequent analyses.

**Current asthma and allergic rhinitis**. Current asthma was defined as mother's report of a doctor's diagnosis of asthma since birth reported on the early teen questionnaire plus report of wheeze or asthma medication in the past year at early teen follow-up (comparison group had no asthma diagnosis, no wheeze, or no asthma medication use). Current allergic rhinitis was defined as mother's report of a doctor's diagnosis of hay fever since birth reported on the early teen questionnaire plus report of sneezing, runny nose, or blocked nose without cold or flu in the past year and current symptoms (moderate-level nasal congestion/stuffiness, nasal blockage, or trouble breathing through the nose in the past month) at the time of nasal swab collection (comparison group had no hay fever diagnosis, no current nasal symptoms at swab collection and no sneezing, runny nose or blocked nose symptoms without cold or flu in the past year).

**IgE and sensitization to environmental allergens**. We measured total IgE by using the ImmunoCAP assay (Phadia, Uppsala, Sweden). Sensitization to environmental allergens was defined as any specific IgE level > 0.35 IU mL$^{-1}$ to common indoor allergens (*Dermatophagoides farinae*, cat and dog dander), mold allergens (*Alternaria* or *Aspergillus* species), or outdoor allergens (rye grass, ragweed, oak, and silver birch).

**Lung function and bronchodilator response**. Trained research assistants measured child height, weight, and lung function.[41] We measured pre-bronchodilator forced expiratory volume in 1 sFEV$_1$ and FVC using the EasyOne Spirometer (NDD Medical Technologies, Andover, MA, USA). We then administered two puffs (90 μg per puff) of albuterol to each participant, and obtained post-bronchodilator spirometry measures at least 15 min after the administration of albuterol. We used the American Thoracic Society criteria for acceptability and reproducibility, with the goal of coaching each subject to produce at least three acceptable and at least two reproducible spirograms. We defined the BDR as a percent change in absolute FEV$_1$ after albuterol administration (postbronchodilator FEV$_1$ – prebronchodilator FEV$_1$/prebronchodilator FEV$_1$ × 100).

**Statistical analyses**. Among 547 participants with high-quality DNAm data eligible for analyses, we report our sample's demographic and biological characteristics using means, standard deviations, or proportions. We performed EWAS CpG-by-CpG by fitting linear regression models using *limma* with moderate test statistics using an empirical Bayes estimation[51]. In EWAS models, we adjusted for variables selected a priori and based on PCA plots: child race/ethnicity, sex, age at sample collection in days, age- and sex-specific BMI z-score using US National Reference Data, smokers currently living in the house, sine and cosine of season of sample collection, maternal education in pregnancy, and cell-type heterogeneity. We performed 10 independent EWAS to analyze nasal DNAm in relation to: (1) current asthma vs. never, (2) current allergic asthma vs. never, (3) FeNO, (4) total serum IgE levels, (5) any environment-specific IgE, (6) FEV z-score, (7) FVC z-score, (8) FEV/FVC z-score, (9) BDR, and (10) current allergic rhinitis vs. never. We bioinformatically controlled for cell-type composition using ReFACTor, a reference-free method to adjust for cell-type composition in genome-wide DNAm studies from heterogenous tissues[52]. We chose ReFACTor as it has been shown to control the false-positive rate even when compared to reference-based methods[36]. We adjusted for the first 10 PCs from ReFACTor (Supplementary Figure 3) as proxy for nasal cellular heterogeneity. CpG-by-CpG EWAS results were adjusted for multiple comparisons using a Bonferroni correction ($P < 6.95 \times 10^{-8}$). Quantile–quantile plots for the regression $P$ values were used to visually inspect genomic inflation and we report the genomic inflation factor ($\lambda$) for unadjusted and cell-type-adjusted analyses. We performed regional DNAm analyses using DMRcate[53] to identify DMRs associated with each trait. We defined a significant DMR as a DNAm region ≥5 CpGs and evaluated statistical significance using a

Stouffer FDR-adjusted $P$ <0.05. We evaluated the overlap among differentially methylated CpGs for each EWAS surviving multiple testing adjustment.

Lastly, we estimated DNAm-Age using the Horvath multi-tissue epigenetic age predictor[40]. We used epigenetic age acceleration from the residuals of a linear model regressing DNAm-Age on chronological age as reported in the Horvath's online calculator: (https://dnamage.genetics.ucla.edu/). All statistical tests were two sided and analyses were carried out using R, version 3.5.0 (www.r-project.org/).

**Replication in epithelial nasal cells**. We sought to replicate our top differentially methylated findings of asthma and allergic asthma in an external cohort with nasal epithelial cells collected from the posterior portion of the inferior turbinate from the Inner City Asthma Consortium[6]. Briefly, in this study samples of nasal epithelial cells from 36 atopic asthmatics and 36 controls with at least 80% ciliated epithelial cells were collected and DNAm was measured using Illumina's Infinium Human Methylation 450K BeadChip. We downloaded publicly available data from the Gene Expression Omnibus repository (GSE65163)[6]. To allow for direct comparability, we carried out the same pre-processing and analytical strategy used in our study, including adjusting for cell type using ReFACTor (9 PCs). Among differentially methylated CpGs found for asthma and allergic asthma, we compared differences in DNAm in adjusted models and controlling the FDR <0.05.

We further tested for replication using a second independent study from The Epigenetic Variation and Childhood Asthma in Puerto Ricans (EVA-PR), a case–control study of childhood asthma in Puerto Rico[22]. Briefly, in this study, nasal epithelial samples from 483 participants aged 9–20 years were collected, and DNAm was measured using the Illumina's Infinium Human Methylation 450K BeadChip. Atopy was defined as at least one positive IgE to five common aeroallergens in Puerto Rico; asthma was defined as physician's diagnosis plus at least one episode of wheezing in the previous year. We used data from the EVA-PR EWAS for atopic asthma (vs. non-atopic controls) as replication for our analyses on asthma and atopic asthma, and the EVA-PR EWAS for atopy as replication for our analysis of IgE sensitization. We adjusted the FDR <0.05 among CpGs found in both analyses.

**Reporting summary**. Further information on research design is available in the Nature Research Reporting Summary linked to this article.

## Data availability
Datasets generated and analyzed during the current study are not publicly available because we did not obtain consent for such public release of epigenetic data from participants. However, raw data to generate figures and tables are available from the corresponding author with the appropriate permission from the Project Viva study team and investigators (project_viva@hphc.org) upon reasonable request and institutional review board approval. Summary statistics for all EWAS performed are available via https://doi.org/10.6084/m9.figshare.8285612.v1.

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

## Acknowledgements

This research was supported by the National Institutes of Health grants R01 AI102960, R01 034568, UG3 OD023286, P30ES000002, R01HL111108, P01 HL132825, HL P01114501, R00HL109162, R01 HL117191, and MD011764.

## Author contributions

All authors read and approved this manuscript. A.C. and J.E.S were the primary authors and data analysts, and contributing equally under the supervision of E.O. and D.R.G. The following authors: S.L.R.-S., W.C., L.L., B.A.C., M.-F.H., P.S.L., A.A.L., K.J.B., D.L.D. and A.A.B. helped conceptualize the study and with the collection of samples, as well as data analyses and drafting of the manuscript. Both E.F. and J.C.C. contributed to the external replication efforts and discussion of results.
