## [Peer Review File · Nature Communications]

Reviewers' comments:

Reviewer #1 (Remarks to the Author):

Review comments for Nature Communications, NCOMMS-18-27162, September 2018

The authors report an EWAS on 547 subjects from the Project Viva study, where associations between nasal methylation levels and different allergy- and asthma-related outcomes were assessed. While the results are intriguing and potentially of great interest, I have a couple of major concerns as outlined below.

Major comments

1) The most obvious limitation in this study is the lack of an independent replication dataset for the top findings. After all, the study includes for example only 65 current asthmatics and 47 rhinitis subjects. Although the authors checked if the Xu et al CpGs for asthma also were significant in this study using nasal epithelial cells, which they were, there is no other way forward than to replicate your own top hits in an independent dataset. As discussed further below, there is reason to believe that there are numerous false positive findings in the unadjusted models in this study. Similar to any state-of-the-art GWAS study of today, where replication attempts are always performed (at least in decent papers), this should be the standard also for EWAS.

2) Given the huge effect that adjustment for estimated cell counts had on the results, it does not make sense to present table after table with top hits not adjusted for cell type (Tables 3-8). The unadjusted models seem to primarily represent cell type composition, which the authors also acknowledge, but these are not the results of interest when it comes to methylation changes and health outcomes. I would strongly recommend presenting cell-type adjusted top findings.

3) Please provide Q-Q plots for the different models, both the unadjusted (for cell type) and the adjusted ones. Looking at the numbers of significant hits in the unadjusted models together with the lambda values suggest issues with inflation and false positive findings. The Manhattan plots also indicate very unspecific patterns, mainly driven by cell-type issues (I would guess). Lambdas from the adjusted models using ReFACTor look much more sensible and trustworthy.

4) Biologically relevant genes were identified for the studied outcomes. There is however yet another limitation in the study in that the authors were not able to directly assess the functional relevance of hypo- or hypermethylation at the identified CpG sites.

5) The use of multiple, related phenotypes could be an asset, but there is also a risk of "fishing expedition" issues and multiple testing problems. Again, the only way to tackle potential drawbacks with such approach is to apply rigorous statistical and methodological analyses and to replicate key findings.

Reviewer #2 (Remarks to the Author):

This is a very interesting study from a leading group. They have, with great care, recovered cells from the anterior nasal airway by swabs and performed an EWAS on extracted DNA. They discover multiple hits that are very relevant to asthma and to atopy.

It is very difficult to fault the methodology of the study, which seems to have been carried out with

great attention to detail at every step, including the elegant statistical analyses. This alone makes the paper important, and establishes the relevance of nasal sampling to the understanding of atopic asthma at the same time as laying down the standards to which all subsequent investigators will refer.

The method for controlling for cell counts appears effective, but has the disadvantage of giving no insights into the relative contribution of specific cell types (eosinophils, neutrophils, etc.) or their functional state. I wonder if the paper could be strengthened by including information from published cell-specific methylation patterns. The use of correlation networks (e.g. from the WGCNA package) may derive cell-specific eigenvectors that can be included in the overall analyses.

It might also be valuable to explore factors that are specific to particular phenotypes such as FeNO, and I wonder if random forest or other analyses may help in this regard.

Reviewer #3 (Remarks to the Author):

This manuscript by Cardenas et al reports results of an epigenome-wide association study (EWAS) of current asthma, allergic sensitization, allergic rhinitis, fractional exhaled nitric oxide (FeNO) and lung function using Illumina EPIC array on nasal swabs from the anterior nares of 547 children from Project Viva (mean-age 12.9y; 67.1% White, 16.1% Black, 4.2 % Hispanic, 3.1% Asian and 9.3% of more than one race). The analysis was done using linear models adjusting for race/ethnicity, sex, age at nasal sample collection, body mass index (BMI) z-score, maternal education, smokers living in the household, and sine and cosine of season at sample collection. The authors also models adjusted for cell proportions estimated by reference-free method, which substantially reduced the number of associations, but they discuss both sets of results. All phenotypes tested have some associations, except for lung function I which case nothing was significant. Validation was performed using data from peripheral blood from Liang et al Nature publication (3 independent cohorts that focused on IgE analysis).

Strengths of the study are in large numbers using a very well established cohort and in superb statistical analysis. There are, however, several limitations of this study as outlined below.

Major issues:

1. Use of cells collected from anterior nares without any measures of cell composition is a major issue with this study that unfortunately cannot be overcome since this information was not collected. I believe this is a HUGE confounder in this analysis. There is no way that these samples are only 1% immune cells if they see so many changes in immune genes and they report replication in peripheral blood. These samples also likely have varying proportions of columnar and squamous epithelia. This cannot be adequately adjusted for with a reference-free method. Samples from inferior turbinates can also have a mix of cells but previously published studies selected samples that predominately have columnar cells. While use of nasal swab from anterior nares may be desirable in younger kids, I am not sure why authors chose this method. They have increased their sample size over previous studies but introduced so much variability and such confounding that they did not try to deal with adequately by obtaining cell count information.
2. There is no gene expression data so there is no way to know which of these methylation changes may have potential functional impact. I am surprised that they do not have gene expression data given the publication they cite to justify the use of this collection method (J Allergy Clin Immunol. 2015 Oct; 136(4): 1120-3.e4) has expression data but this manuscript says that the authors "were unable to collect sufficient high-quality RNA from the anterior nares".
3. While there is replication in blood, there is no attempt to replicate in another cohort of nasal

epithelial cells, and yet conclusions are drawn about nasal epithelia.

4. The way results are presented is extremely confusing. They report single CpGs and regions, with and without adjustment for cell proportions, in relation to 4 outcomes: current asthma, allergic sensitization, allergic rhinitis, and FeNO. The summary table is helpful but results are discussed jumping from one model to another, and it is hard to follow. Are there any commonalities among all these models? Or maybe pick one outcomes to focus discussion on?

Minor comments:

1. Are there differences in methylation by race/ethnicity? Main models adjusted for it but it seems that putting all these diverse individuals may wash out some signal specific to one race/ethnicity.
2. First ICAC study (97 allergic asthmatics vs 97 controls) was in PBMCs and not in nasal cells (the second study of 36 allergic asthmatics vs 36 controls was in nasal cells).

Reviewer #1 (Remarks to the Author):

The authors report an EWAS on 547 subjects from the Project Viva study, where associations between nasal methylation levels and different allergy- and asthma-related outcomes were assessed. While the results are intriguing and potentially of great interest, I have a couple of major concerns as outlined below.

Major comments

1) The most obvious limitation in this study is the lack of an independent replication dataset for the top findings. After all, the study includes for example only 65 current asthmatics and 47 rhinitis subjects. Although the authors checked if the Xu et al CpGs for asthma also were significant in this study using nasal epithelial cells, which they were, there is no other way forward than to replicate your own top hits in an independent dataset. As discussed further below, there is reason to believe that there are numerous false positive findings in the unadjusted models in this study. Similar to any state-of-the-art GWAS study of today, where replication attempts are always performed (at least in decent papers), this should be the standard also for EWAS.

Response: We thank the reviewer for this constructive comment and completely agree that replication should be the standard. This suggestion motivated us to look for external replication of our results. We have majorly revised our manuscript and results to include **replication** of findings in nasal epithelial cells of an independent cohort of children. We now only present and interpret results of analyses fully adjusted for cell-type heterogeneity. We incorporated the following major revisions based on the reviewer's comments:

- 1) We excluded all unadjusted findings and only discussed, present, and interpret cell-type adjusted results throughout the manuscript to reduce the potential for false positives as suggested.
- 2) Additionally, we used an external replication cohort of inner-city children aged 10 to 12 years with persistent atopic asthma (n=36) versus healthy control subjects (n=36) with nasal epithelial DNA methylation measurements (Yang, Ivana V., et al. *Journal of Allergy and Clinical Immunology* 139.5 (2017): 1478-1488; PMID: 27745942). The largest limitation of this replication cohort is that samples were measured using Illumina's Infinium Human Methylation 450k BeadChip sharing approximately 328,000 CpGs with the newer EPIC BeadChip used in our study. In this replication cohort, nasal sampling occurred in the inferior turbinate in contrast to the anterior nares location from our study. However, this is currently the most similar dataset publically available to attempt replication for our multiple findings and provide comparability to the published literature.

To test for external replication, we downloaded publicly available data from the Gene Expression Omnibus GEO (GSE65163) from the study performed by Yang and colleagues. We now describe this replication approach in the manuscript with track changes on **page 25 lines 527-538:**

“Replication in Epithelial Nasal Cells. We sought to replicate our top differentially methylated findings of asthma and allergic asthma in an external cohort with nasal epithelial cells collected from the posterior portion of the inferior turbinate from the Inner City Asthma Consortium¹. Briefly, in this study samples of nasal epithelial cells from 36 atopic asthmatic and 36 controls with at least 80% ciliated epithelial cells were collected and DNA methylation was measured using Illumina's Infinium Human Methylation 450K BeadChip. We downloaded publicly available data from the Gene Expression Omnibus repository (GSE65163) from the study performed by Yang and colleagues¹. To allow for direct comparability we carried out the same pre-processing and analytical strategy used in our study, including adjusting for cell-type using ReFACTor (9 PCs). Among differentially methylated CpGs found for asthma and allergic asthma we compared differences in DNA methylation in adjusted models and controlling the FDR<0.05.”

Although this replication cohort was relatively small (36 kids with atopic asthma and 36 controls) we were able to replicate many of our findings after adjusting for multiple comparisons in the replication phase. Namely, among the 95 CpGs found to be differentially methylated relative to current asthma in our data and present in the 450K assay of Yang and colleagues, 58 CpGs replicated (61%) with FDR<0.05 and relative small unadjusted p-values ($p < 1 \times 10^{-4}$) in the replication cohort with the same direction and similar magnitude of association for all 58 CpGs.

For allergic asthma 375 CpGs were present in the Yang *et al.* 450K data of which 199 CpGs (53%) replicated with an FDR<0.05 and 197 of these had consistent direction for the association.

Table R1. Replication from the independent cohort with nasal 450K data of children with atopic asthma

EWAS	CpGs findings from our study (850K/EPIC Chip)	Present in Yang et al. (450K/EPIC Chip)	Replicated FDR<0.05
Current Asthma vs. never	285	95	58 (61%)
Allergic asthma vs. never	1,235	375	199 (53%)

We now include the results from this replication on **pages 10-11 lines 212-225**:

“Replication in Epithelial Nasal Cells. We tested for replication of our top differentially methylated findings of asthma and allergic asthma in an external cohort with nasal epithelial cells collected from the posterior portion of the inferior turbinate and analyzed with the Infinium HumanMethylation450K BeadChip⁶. We checked for replication of the 285 differentially methylated CpGs for asthmatics of which 95 probes were present on the 450K from the replication cohort. Among the 95 CpGs found to be differentially methylated for asthmatics in our study and present in the 450K, 58 CpGs replicated (61%) after controlling for multiple comparisons (FDR<0.05 for 95 comparisons) in the replication cohort with consistent direction and magnitude of association for all 58 CpGs (**Supplemental Table 10**). Replicated CpGs included several sites annotated to the ACOT7, EPX and EVL genes. For allergic asthma analyses there were 375 CpG probes present in the Yang *et al.* 450K dataset of which 199 CpGs (53%) replicated with an FDR<0.05 and 197 of these had consistent direction of association (**Supplemental Table**

11). Replicated CpGs included several sites annotated to the ACOT7, ZFPM1, PRG2, EPX and EVL genes.”

2) Given the huge effect that adjustment for estimated cell counts had on the results, it does not make sense to present table after table with top hits not adjusted for cell type (Tables 3-8). The unadjusted models seem to primarily represent cell type composition, which the authors also acknowledge, but these are not the results of interest when it comes to methylation changes and health outcomes. I would strongly recommend presenting cell-type adjusted top findings.

Response: This is a valid concern and valuable comment that motivated us to re-evaluate our approach and presentation of results. We followed the suggestion and removed all unadjusted analyses (Tables 3-8). We now only present tables for top differentially methylated CpGs and regions found in analyses bioinformatically adjusted for cell-type (Table 2) presented below:

Table 2. Summary of nasal Epigenome-Wide Association Analyses (EWAS): number of differentially methylated CpGs (FDR<0.05 and Bonferroni correction) and Differentially Methylated Regions (DMRs; FDR<0.05).

EWAS*	Sample size (N; n=cases)	λ	*Fully Adjusted EWAS		
			FDR<0.05 CpGs	Bonferroni CpGs	DMRs
Current Asthma vs. never	N=463; n=65	0.91	285	1	0
Fractional exhaled nitric oxide (FeNO)	N=517	1.32	8,372	744	191
Current allergic rhinitis vs. never	N=277; n=47	1.08	0	0	0
Total IgE	N=361	1.09	3	1	3
Environment IgE sensitization	N=361; n=211	0.93	17	6	4
Current asthma with environment IgE sensitization	N=301; n=36	1.00	1,235	15	7
FEV z-score	N=525	0.99	0	0	0
FVC z-score	N=525	0.95	0	0	0
FEV/FVC z-score	N=525	1.11	0	0	0
%-Bronchodilator response	N=454	0.87	130	50	0

*Analyses adjusted for child race/ethnicity, sex, age at sample collection, BMI z-score, maternal education, smokers living in the house, sine and cosine of season at sample collection and cellular heterogeneity (10 PCs from ReFACToR).

λ =genomic inflation factor

DMRs: differentially methylated regions with ≥ 5 CpGs with a Stouffer FDR $P < 0.05$.

Finally, we include all fully adjusted results, both CpGs and regions, in a supplement csv file for future reproducibility and comparison (**Supplementary Tables S3-S9**).

3) Please provide Q-Q plots for the different models, both the unadjusted (for cell type) and the adjusted ones. Looking at the numbers of significant hits in the unadjusted models together with the lambda values suggest issues with inflation and false positive findings. The Manhattan plots also indicate very unspecific patterns, mainly driven by cell-type issues (I would guess). Lambdas from the adjusted models using ReFACTor look much more sensible and trustworthy.

Response: This is a great suggestion. We now provide overlapping Q-Q plots for models adjusted and not adjusted for cell-type heterogeneity for all the 10-EWAS performed in the supplementary material (**Supplementary Figure S1**) shown below. As predicted by the reviewer, unadjusted results show very large inflation (black). Furthermore, we now only present Manhattan plots adjusted for cell-type composition in **Figure 2**, which as predicted are greatly attenuated.

We now include this information on the manuscript **page 6 lines 109-112**:

“Genomic inflation observed in EWAS not adjusted for cell-type heterogeneity was greatly attenuated after adjustment for surrogates of cell-types estimated via ReFACTor (Supplementary Figure S1). Manhattan plots of fully adjusted EWAS of FeNO, current asthma and allergic asthma are shown in Figure 2.

Supplementary Figure S1. Quantile-Quantile plots of expected vs observed p-values from epigenome-wide association analyses (EWAS): p-values adjusted for confounders but no cell-type are in black and p-values from models further adjusted for cell-type heterogeneity using ReFACTor (10 PCs) are in blue.

4) Biologically relevant genes were identified for the studied outcomes. There is however yet another limitation in the study in that the authors were not able to directly assess the functional relevance of hypo- or hypermethylation at the identified CpG sites.

Response: This is correct and a limitation of this study and was likely a consequence of sampling in the anterior nares, which results in adequate DNA with comparable methylation measures to inferior turbinate samples, but may result in insufficient RNA for gene expression measurement (Lai, Peggy S., et al. "Alternate methods of nasal epithelial cell sampling for airway genomic studies." *Journal of Allergy and Clinical Immunology* 136.4 (2015): 1120-1123; PMID: [26037550](https://pubmed.ncbi.nlm.nih.gov/26037550/)). For us, the great advantage to sampling in this location was the acceptability of the procedure to the children (also documented by Lai 2015) and to the Institutional Review Board (IRB) that approved the protocols, with the resultant large numbers of children providing samples. Sampling from the inferior involves using a speculum and greater discomfort. Thus, while we were able to obtain sufficient DNA for high quality DNA methylation assessment, unfortunately, we were not able to obtain a sufficient RNA and/or of high enough quality perform transcriptomic analyses. For example, out of 250 samples for which we attempted to recover RNA half had concentration lower than 10 ng/uL and not a single samples had as much as 250 ng/uL. In addition, preliminary quality checks using spectrophotometry showed low OD ratios for most samples (A260/A280 in usually lower than 2.0). Based on these disappointing initial results, we elected to abort the RNA analysis and we did not proceed to measure RIN values nor to conduct any transcriptomics analysis. We now state this on the manuscript **pages 18-19 lines 395-397:**

“Additionally, while it was our intention to measure gene expression we were unable to collect sufficient high-quality RNA from the anterior nares or enough concentration (most samples yielded less than 10 ng/uL).”

We also stated our rationale and justification for sampling from the anterior nares on **page 17 lines 357-359**:

“Sampling from the anterior nares does not require a speculum, and subjects report less discomfort with this technique as compared to inferior turbinate sampling, making it more amenable for use in pediatric populations³⁹”.

5) The use of multiple, related phenotypes could be an asset, but there is also a risk of “fishing expedition” issues and multiple testing problems. Again, the only way to tackle potential drawbacks with such approach is to apply rigorous statistical and methodological analyses and to replicate key findings.

Response: We chose to present all analyses conducted for transparency. As mentioned the IgE, asthma and allergic asthma results are multiple related phenotypes, but we believe presenting all results will allow testing for reproducibility in future studies, which might collect one outcome but not the other. In addition, we hypothesized that lung function would be related to nasal epithelial DNA methylation, but this hypothesis was not supported. We would still like include these results, as it could be very important for other cohorts in the near future that might want to test this hypothesis.

Lastly, our replication efforts show generalizability of findings even for samples from other regions of the nose (*i.e.* inferior turbinate), different ethnic population and DNA methylation technology (450K). Those we believe that your results are generalizable.

Reviewer #2 (Remarks to the Author):

This is a very interesting study from a leading group. They have, with great care, recovered cells from the anterior nasal airway by swabs and performed an EWAS on extracted DNA. They discover multiple hits that are very relevant to asthma and to atopy.

It is very difficult to fault the methodology of the study, which seems to have been carried out with great attention to detail at every step, including the elegant statistical analyses. This alone makes the paper important, and establishes the relevance of nasal sampling to the understanding of atopic asthma at the same time as laying down the standards to which all subsequent investigators will refer.

The method for controlling for cell counts appears effective, but has the disadvantage of giving no insights into the relative contribution of specific cell types (eosinophils, neutrophils, etc.) or their functional state. I wonder if the paper could be strengthened by including information from published cell-specific methylation patterns. The use of correlation networks (e.g. from the WGCNA package) may derive cell-specific eigenvectors that can be included in the overall analyses.

It might also be valuable to explore factors that are specific to particular phenotypes such as FeNO, and I wonder if random forest or other analyses may help in this regard.

Response: We appreciate the reviewer's positive comments. The suggestions above motivated us to do another analysis in regards to factors that are specific to some outcomes. To improve interpretability we now performed enrichment analyses of gene ontology categories for allergic asthma, asthma and FeNO findings fully adjusted for cell-type (**Figure 3**) shown below:

We find overlap for differentially methylated genes among allergic asthmatics and FeNO measurements for the neutrophil degranulation, signaling by interleukins and interleukin-4 and interleukin-13 signaling biological processes. We now elaborate on this on **page 10 lines 192-196**:

“Among differentially methylated genes, we also looked at biological pathways enriched in the Reactome database. The top differentially methylated Reactome pathway was observed for neutrophil degranulation followed by signaling by interleukins pathways for allergic asthma and FeNO. Genes found for asthmatics were enriched in the interleukin-2 family signaling as well as the SUMOylation of intracellular receptor pathways (Figure 3).”

Reviewer #3 (Remarks to the Author):

This manuscript by Cardenas et al reports results of an epigenome-wide association study (EWAS) of current asthma, allergic sensitization, allergic rhinitis, fractional exhaled nitric oxide (FeNO) and lung function using Illumina EPIC array on nasal swabs from the anterior nares of 547 children from Project Viva (mean-age 12.9y; 67.1% White, 16.1% Black, 4.2 % Hispanic, 3.1% Asian and 9.3% of more than one race). The analysis was done using linear models adjusting for race/ethnicity, sex, age at nasal sample collection, body mass index (BMI) z-score, maternal education, smokers living in the household, and sine and cosine of season at sample collection. The authors also models adjusted for cell proportions estimated by reference-free method, which substantially reduced the number of associations, but they discuss both sets of results. All phenotypes tested have some associations, except for lung function I which case nothing was significant. Validation was performed using data from peripheral blood from Liang et al Nature publication (3 independent cohorts that focused on IgE analysis).

Strengths of the study are in large numbers using a very well established cohort and in superb statistical analysis. There are, however, several limitations of this study as outlined below.

Major issues:

1. Use of cells collected from anterior nares without any measures of cell composition is a major issue with this study that unfortunately cannot be overcome since this information was not collected. I believe this is a HUGE confounder in this analysis. There is no way that these samples are only 1% immune cells if they see so many changes in immune genes and they report replication in peripheral blood. These samples also likely have varying proportions of columnar and squamous epithelia. This cannot be adequately adjusted for with a reference-free method. Samples from inferior turbinates can also have a mix of cells but previously published studies selected samples that predominately have columnar cells. While use of nasal swab from anterior nares may be desirable in younger kids, I am not sure why authors chose this method. They have increased their sample size over previous studies but introduced so much variability and such confounding that they did not try to deal with adequately by obtaining cell count information.

Response: We appreciate the reviewer's concern and agree that even samples from the inferior turbinates contain a mixture of cells. Therefore, in theory there is no ideal sample. To ensure generalizability of our findings, we replicated our results in an independent cohort of nasal DNA methylation samples collected from asthmatics and controls from posterior portion of the inferior turbinate, verified to have at least 80% ciliated epithelial cells visualized from slides (Yang, Ivana V., et al. *Journal of Allergy and Clinical Immunology* 139.5 (2017): 1478-1488; PMID: 27745942). We replicated over 50% of our findings present in their data for asthma and allergic asthma even after adjusting for multiple comparisons in the replication phase. We believe this is evidence that our samples are appropriate and comparable to those that might be obtained from the inferior turbinates.

We now include the results from this replication on **pages 10-11 lines 212-225**:

“Replication in Epithelial Nasal Cells. We tested for replication of our top differentially methylated findings of asthma and allergic asthma in an external cohort with nasal epithelial cells collected from the posterior portion of the inferior turbinate and analyzed with the Infinium HumanMethylation450K BeadChip⁶. We checked for replication of the 285 differentially methylated CpGs for asthmatics of which 95 probes were present on the 450K from the replication cohort. Among the 95 CpGs found to be differentially methylated for asthmatics in our study and present in the 450K, 58 CpGs replicated (61%) after controlling for multiple comparisons (FDR<0.05 for 95 comparisons) in the replication cohort with consistent direction and magnitude of association for all 58 CpGs (Supplemental Table 10). Replicated CpGs included several sites annotated to the ACOT7, EPX and EVL genes. For allergic asthma analyses there were 375 CpG probes present in the Yang et al. 450K dataset of which 199 CpGs (53%) replicated with an FDR<0.05 and 197 of these had consistent direction of association (Supplemental Table 11). Replicated CpGs included several sites annotated to the ACOT7, ZFPM1, PRG2, EPX and EVL genes.”

In terms of confounding by cell-type, the reference free method we used (ReFACTor) has been shown to adequately control the false positive rate even in highly confounded scenarios. For example, in highly confounded epigenome-wide association analyses of DNAm from mixed leukocytes of rheumatoid arthritis cases and controls (ReFACToR: Rahmani, Elior, et al. *Nature Methods* 13.5 (2016): 443; PMID: 27018579). This is a limitation for all studies, including blood-based epigenome-wide association analyses. Even analyses adjusted for leukocyte composition have been shown to reflect an expansion of rare cell sub-types that cannot be captured with standard methods (Bauer, Mario, et al. "A varying T cell subtype explains apparent tobacco smoking induced single CpG hypomethylation in whole blood." *Clinical Epigenetics* 7.1 (2015): 81; PMID: 26246861).

However, the method we applied (ReFACTor) has been proven to adequately control the false discovery rate as stated on **page 24 lines 511-512**:

“We chose ReFACTor as it has been shown to control the false positive rate even when compared to reference-based methods⁴⁰.”

⁴⁰ Rahmani, Elior, et al. "Correcting for cell-type heterogeneity in DNA methylation: a comprehensive evaluation." *Nature methods* 14.3 (2017): PMID: 28245214

2. There is no gene expression data so there is no way to know which of these methylation changes may have potential functional impact. I am surprised that they do not have gene expression data given the publication they cite to justify the use of this collection method (J Allergy Clin Immunol. 2015 Oct;136(4):1120-3.e4) has expression data but this manuscript says that the authors “were unable to collect sufficient high-quality RNA from the anterior nares”.

Response: This is correct, the study referenced sampled from both the anterior nares and also the inferior turbinate from the same participants to compare DNA methylation and expression. The study found high correlation for DNA methylation in both sampling locations. However, that manuscript observed significantly lower RNA integrity number (RIN) in the inferior turbinate compared to the anterior nares (8.9 vs 2.2 $p < 0.001$) the referenced study concludes:

“RNA quantity and degradation in anterior nares samples likely limit the use of this method for expression studies”

For example, out of 250 samples for which we attempted to recover RNA half had concentration lower than 10 ng/uL and not a single samples had as much as 250 ng/uL. Also, preliminary quality checks using spectrophotometry showed low OD ratios for most samples (A260/A280 in usually lower than 2.0). Based on these disappointing initial results, we elected to abort the RNA analysis and we did not proceed to measure RIN values nor to conduct any transcriptomics analysis. We now state this on the manuscript **page 18 lines 395-397**:

“Additionally, while it was our intention to measure gene expression we were unable to collect sufficient high-quality RNA from the anterior nares or enough concentration (most samples yielded less than 10 ng/uL).”

3. While these is replication in blood, there is no attempt to replicate in another cohort of nasal epithelial cells, and yet conclusions are drawn about nasal epithelia.

Response: This suggestion motivated us to look for replication studies. To test for external replication, we downloaded publicly available data from the Gene Expression Omnibus GEO (GSE65163) from the study performed by Yang and colleagues on nasal cells of asthmatics and controls. (Yang, Ivana V., et al. Journal of Allergy and Clinical Immunology 139.5 (2017): 1478-1488; PMID: 27745942).

We now include the results from the replication phase on this publicly available dataset on **pages 10-11 lines 212-225**:

“Replication in Epithelial Nasal Cells. We tested for replication of our top differentially methylated findings of asthma and allergic asthma in an external cohort with nasal epithelial cells collected from the posterior portion of the inferior turbinate and analyzed with the Infinium HumanMethylation450K BeadChip¹. We checked for replication of the 285 differentially methylated CpGs for asthmatics of which 95 probes were found in the 450K from the replication cohort. Among the 95 CpGs found to be differentially methylated for asthmatics in our study and present in the 450K, 58 CpGs replicated (61%) after controlling for multiple comparisons ($FDR < 0.05$ for 95 comparisons) in the replication cohort with consistent direction and magnitude of association for all 58 CpGs (Supplemental Table 10). Replicated CpGs included several sites annotated to the ACOT7, EPX and EVL genes. For allergic asthma analyses there were 375 CpG probes present in the Yang et al. 450K dataset of which 199 CpGs (53%) replicated with an $FDR < 0.05$ and 197 of these had consistent direction of association (Supplemental Table

S11). Replicated CpGs included several sites annotated to the ACOT7, ZFP1, PRG2, EPX and EVL genes.”

Several of the genes for which we observed multiple differentially methylated CpGs also replicated on this independent cohort that used a different array (450K) and samples from a different region (inferior turbinate). Therefore, we are confident that replicated sites are robust to population and sampling location/technique.

4. The way results are presented is extremely confusing. They report single CpGs and regions, with and without adjustment for cell proportions, in relation to 4 outcomes: current asthma, allergic sensitization, allergic rhinitis, and FeNO. The summary table is helpful but results are discussed jumping from one model to another, and it is hard to follow. Are there any commonalities among all these models? Or maybe pick one outcomes to focus discussion on?

Response: We thank the reviewer for this suggestion we now only discuss cell-type adjusted analyses throughout the manuscript.

Minor comments:

1. Are there differences in methylation by race/ethnicity? Main models adjusted for it but it seems that putting all these diverse individuals may wash out some signal specific to one race/ethnicity.

Response: We fully agree with this comment. However, given the relative small number of individuals with of race/ethnicity other than white we are not able to perform the suggested analyses. For example the second most frequent category is black (n=88) for which only 13 suffered from asthma.

2. First ICAC study (97 allergic asthmatics vs 97 controls) was in PBMCs and not in nasal cells (the second study of 36 allergic asthmatics vs 36 controls was in nasal cells).

Response: Thank you for this observation we corrected this on **page 15 lines 331-343**.

Reviewers' comments:

Reviewer #1 (Remarks to the Author):

Review comments on the revised version, NCOMMS-18-27162A

The authors have addressed my major concerns, and I am very glad to see a replication dataset added, and cell-type adjustments included. The new QQ plots really show the cell-type influence on the results, which is also an important finding.

It would have been very informative to see methylation - expression correlations in nasal epithelial cells, but I understand the challenge to get reliable data.

Reviewer #2 (Remarks to the Author):

I still think this is an exemplary study of the nasal methylome. The authors have addressed all of my concerns, and the responses to input from the other referees has significantly improved the manuscript.

Reviewer #3 (Remarks to the Author):

The authors did a very nice job with the revisions in response to reviewer comments. Major issues that have been fixed are (1) focus only on adjusted analyses and show qq plots, and (2) replication in a published dataset.

It is very unfortunate that they have no way of assessing expression in their own samples but this is in part alleviated by some replication (only 50%) of their findings in ICAC, where it was shown that methylation changes also are associated with corresponding gene expression changes.

I hate to suggest this in the second round of revisions but I would encourage replication in the larger dataset of nasal epithelial cells that was published in December by Forno et al. in Lancet Respiratory. They were able to replicate ICAC findings but also identify additional methylation changes with their much larger sample size. This would really help with both replication (hopefully better than 50%) and lack of expression (since they show association of methylation changes with expression). They also showed similar findings in EPCAM+ epithelial cells in a smaller subset of subjects, so this helps alleviate any remaining concerns with cell mixture.

Reviewer #1 (Remarks to the Author):

Review comments on the revised version, NCOMMS-18-27162A

The authors have addressed my major concerns, and I am very glad to see a replication dataset added, and cell-type adjustments included. The new QQ plots really show the cell-type influence on the results, which is also an important finding.

It would have been very informative to see methylation - expression correlations in nasal epithelial cells, but I understand the challenge to get reliable data.

Response: We are thankful for the useful suggestions that improved the overall quality of the manuscript. In addition, we now added results of a second replication study of nasal epithelial cells as suggested by the third reviewer. We again replicated over 50% of the sites common between studies (450K and EPIC) with outstanding consistency of directionality of associations across studies. We also note that several genes and CpGs reported to be associated with atopy in this cohort were also correlated with gene expression and found among top hits for our analyses. See response to the third reviewer.

Reviewer #2 (Remarks to the Author):

I still think this is an exemplary study of the nasal methylome. The authors have addressed all of my concerns, and the responses to input from the other referees has significantly improved the manuscript.

Response: We appreciate all the comments that significantly improved the quality of the manuscript.

Reviewer #3 (Remarks to the Author):

The authors did a very nice job with the revisions in response to reviewer comments. Major issues that have been fixed are (1) focus only on adjusted analyses and show qq plots, and (2) replication in a published dataset.

It is very unfortunate that they have no way of assessing expression in their own samples but this is in part alleviated by some replication (only 50%) of their findings in ICAC, where it was shown that methylation changes also are associated with corresponding gene expression changes.

I hate to suggest this in the second round of revisions but I would encourage replication in the larger dataset of nasal epithelial cells that was published in December by Forno et al. in Lancet Respiratory. They were able to replicate ICAC findings but also identify additional methylation changes with their much larger sample size. This would really help with both replication (hopefully better than 50%) and lack of expression (since they show association of methylation

changes with expression). They also showed similar findings in EPCAM+ epithelial cells in a smaller subset of subjects, so this helps alleviate any remaining concerns with cell mixture.

Response: We thank the reviewer for motivating us to seek replication in a second study of the nasal methylome. We reached out to Dr. Forno and colleagues and now present their results along with ours for CpG sites found in our study and in their 450K nasal epithelial data from The Epigenetic Variation and Childhood Asthma in Puerto Ricans (EVA-PR) case-control study. A limitation is the difference in array coverage between our cohort (EPIC array with 719,075 CpGs after QC) and the EVA-PR study (450K array with 227,836 CpGs after QC).

Dr. Forno and colleagues annotated their results to our findings and we now present replication for three comparisons from our data:

- 1) EWAS results of atopic asthma in EVA-PR compared to asthma in Project Viva (**Table S10**)
- 2) EWAS results of atopic asthma in EVA-PR compared to allergic asthma in Project Viva (**Table S11**)
- 3) EWAS results of atopy in EVA-PR compared to Environment IgE sensitization in Project Viva (**Table S12**)

We now summarize these findings in the revised version of the manuscript on **pages 9-10 lines 187-204**:

*“We also compared our results to data from EVA-PR, a case-control study of asthma in Puerto Rican children and adolescents, which measured the nasal methylome using nasal epithelial cells collected from the inferior turbinate²². In this external cohort’s findings for atopic asthma, we identified 61% (48/79) CpGs that replicated (FDR<0.05 for 48 comparisons) all with consistent direction of association including multiple CpGs annotated to EVL and EPX genes for our asthma results (**Supplemental Table S10**). For our allergic asthma results, 59% (187/315) of CpGs replicated (FDR<0.05 for 187 comparisons) but two had opposite direction of association. Lastly, for environment IgE sensitization 4 out of 7 CpGs replicated (GJA4, ACOX2, PRKAG2, CYP27B1/METTL1) and all 7 had consistent direction of association (**Supplemental Table S12**). Notably, in the EVA-PR study analysis of atopy two of our top CpGs were replicated among the top 30 differentially methylated findings from sorted CD326+ epithelial cells (cg15006973; GJA4 and cg20372759; CYP27B1/METTL1 genes). In this cohort, they were able to test for DNAm and gene expression relationships and several of our top differentially methylated CpGs were associated with expression. Namely, for environment IgE sensitization (METTL1), for allergic asthma (NTRK1) and several for FeNO (MAP3K14; NTRK1; FBXL7; PCSK6; SLC9A3; CDH26; CAPN14 and MAP3K14).”*

Although we were not able to test for gene expression, results from this cohort support that some of the changes are functional and show the robustness of our findings replicating across two cohorts using different methodologies and statistical modeling strategies.

We also added more details of the data and methods used in this cohort, in our methods section on **pages 23-24 lines 498-507**:

“We further tested for replication using a second independent study from The Epigenetic Variation and Childhood Asthma in Puerto Ricans (EVA-PR), a case-control study of childhood asthma in Puerto Rico²². Briefly, in this study, nasal epithelial samples from 483 participants aged 9-20 years were collected, and DNA methylation was measured using the Illumina's Infinium Human Methylation 450K BeadChip. Atopy was defined as at least one positive IgE to five common aeroallergens in Puerto Rico; asthma was defined as physician's diagnosis plus at least one episode of wheezing in the previous year. We used data from the EVA-PR EWAS for atopic asthma (vs non-atopic controls) as replication for our analyses on asthma and atopic asthma; and the EVA-PR EWAS for atopy as replication for our analysis of IgE sensitization. We adjusted the $FDR < 0.05$ among CpGs found in both analyses.”

REVIEWERS' COMMENTS:

Reviewer #3 (Remarks to the Author):

This is an excellent manuscript, with two replciation cohorts now included. Great work!